# DroneDINO: Towards Heterogeneous Routed Mixture of Experts for Drone-based Unified Object Detection

Dongdong Li [1] [*]   Rui Chen [1] [*]   Yan Fan [1]   Yan Liu [1]   Yangliu Kuai [1]   Pengfei Zhu [1] [2]

## Abstract

Recently, the rapid development of low-altitude aerial applications has driven the need for drone-based unified detectors. In contrast to task-specific detectors that suffer from poor scalability across diverse scenarios, existing unified detectors leverage the Mixture-of-Experts (MoE) architecture to learn task-aware features from diverse datasets. However, the imbalanced multi-task data distribution leads to over-activation of experts for dominant tasks and under-activation for others. To enable balanced feature learning, this paper combines three detection paradigms (RGB, IR, and RGB-IR) into a unified framework termed DroneDINO. DroneDINO extends DINO by introducing heterogeneous routed MoEs that organize experts into three functional groups: shared, task-specific, and dynamic. Unlike conventional dynamic experts where the top-$k$ experts are activated for each input, the shared expert is activated for all inputs, while each task-specific expert is activated exclusively for the matching task. To ensure inputs are routed to appropriate experts and yield task-discriminative features, we propose a task-recognition auxiliary training strategy to penalize features with low task-discriminability. Experiments demonstrate the effectiveness and generalizability of DroneDINO, which consistently outperforms state-of-the-art unified and task-specific detectors across multiple drone-based detection benchmarks.

## 1. Introduction

With the rapid advancement of low-altitude intelligent technologies, drones have exhibited enormous application po-

*Equal contribution [1]National University of Defense Technology, Changsha, Hunan, China [2]Tianjin University, Tianjin, China. Correspondence to: Pengfei Zhu <zhupengfei@tju.edu.cn>.

*Proceedings of the 43rd International Conference on Machine Learning*, Seoul, South Korea. PMLR 306, 2026. Copyright 2026 by the author(s).

tential including urban monitoring, intelligent transportation and logistics distribution. In these applications (Zhai et al., 2023), efficient and accurate object detection serves as the cornerstone technology for drone-based intelligent perception capabilities (Girshick, 2015). The successive emergence of multi-task datasets and task-specific detectors have greatly advanced progress in this field (Zhu et al., 2021; Zhou et al., 2022).

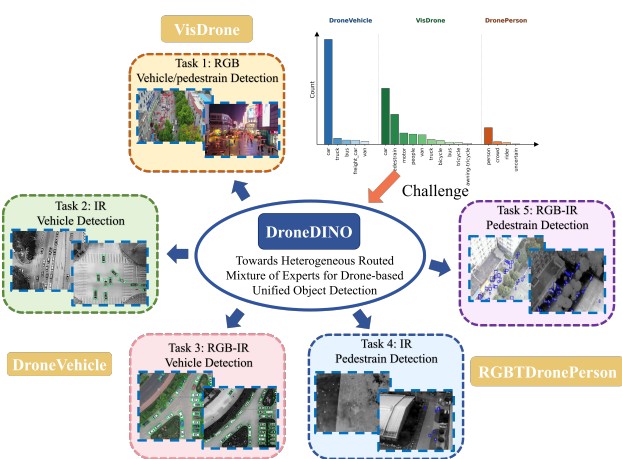

*Figure 1.* We present a novel Drone-UOD task and its baseline DroneDINO, unifying five subtasks across three drone-based datasets in a single training session. To address the cross-dataset distribution imbalance and long-tail effects (see the top-right), we propose a synergy of Task-Recognition Strategy and Heterogeneous Routed Mixture of Experts (HR-MoE), which enables the model to adaptively route features across heterogeneous distributions.

Mainstream drone-based datasets (e.g., VisDrone (Du et al., 2019), DroneVehicle (Sun et al., 2022), RGBTDronePerson (Zhang et al., 2023b)) exhibit significant heterogeneity in imaging altitude, viewing angle, background complexity, image modality and annotation label space (Lee, 2021). Traditional task-specific detectors are designed exclusively for a single specific dataset and suffer from two primary limitations. **First**, these detectors exhibit poor cross-task generalization capability, as inherent dataset heterogeneity compels researchers to train specialized models tailored to each specific dataset, resulting in severe model redundancy and deployment challenges. **Second**, current approaches

lack flexibility and adaptability in handling diverse input formats. To achieve all-weather and all-scene perception, modern drones are often equipped with both RGB and IR sensors (Zeng et al., 2024; Bao et al., 2023) to simultaneously capture images from multiple modalities (Yuan et al., 2022; Yuan & Wei, 2024). However, existing task-specific detectors feature a fixed input interface that handles only one input format (i.e., RGB, IR or RGB-IR).

Recent unified detectors (Li et al., 2024; Zhang et al., 2025b) demonstrate strong generalization in learning shared knowledge across diverse datasets within a shared backbone. They aim to construct a unified foundation model for joint training across all datasets, regardless of heterogeneous label space and modalities. To bridge task gaps while preserving task-specific representation capacity, these approaches integrate Mixture-of-Experts (MoE) into backbones to adaptively process heterogeneous inputs with dynamic routing. However, due to the inherent imbalanced data distribution stemming from different tasks or the long-tail effects(see figure 1), experts associated with the primary modality or head classes are over-activated while other experts for the auxiliary modality and tail classes are under-activated. This imbalanced expert activation impedes balanced learning of shared and task-specific knowledge across different datasets, limiting generalization.

To address the above challenges, this paper defines a novel research task, Drone-Based Unified Object Detection (Drone-UOD). This task requires constructing a single unified model performing detection across heterogeneous datasets and flexibly processing three input formats: visible (RGB), infrared (IR), and visible-infrared fusion (RGB-IR). Guided by the available modality and annotation label space of mainstream Drone datasets (Du et al., 2019; Sun et al., 2022; Zhang et al., 2023b), we unify these datasets into five representative downstream tasks, as illustrated in Figure 1. Based on this new task, we propose a baseline model dubbed DroneDINO rooted in the DINO framework (Zhang et al., 2022). DroneDINO addresses the challenge of imbalanced feature learning from two perspectives: the routing principle of MoEs and model optimization strategy:

**Routing Principle:** Conventional MoEs route input tokens to the top-k activated experts based on the same principle regardless of the relative importance of different tokens. This homogeneous routing principle results in over-activation of dominant experts and under-activation of other experts. To mitigate this issue, we propose to integrate a plug-and-play Heterogeneous Routed Mixture of Experts (HR-MoE) module into the DINO backbone. Specifically, HR-MoE organizes experts into the shared, task-specific and dynamic groups and routes input tokens based on heterogeneous principles for each group.

**Model Optimization:** Existing unified detectors adopt an open-loop feature learning strategy characterized by the lack of effective feedback mechanism. This flawed strategy leads to misalignment between encoded feature and the corresponding input task. To ensure generated features strictly align with their expected task, we propose a task-recognition auxiliary training strategy that penalize low-discriminability features.

The main contributions of this paper are summarized as follows:

- We formulate a new Drone-UOD task and propose its baseline framework dubbed DroneDINO, which can flexibly handle multiple input formats from heterogeneous datasets. Experiments validate DroneDINO consistently outperforms state-of-the-art unified and task-specific detectors across VisDrone, DroneVehicle and RGBTDronePerson.

- We revise the traditional homogeneous routing principle of standard MoEs by proposing the HR-MoE module which ensures balanced expert activation across diverse tasks.

- We introduce a task-recognition auxiliary training strategy to align encoded feature with the input task and enhancing task-specific discriminability of feature representation.

## 2. Related Work

### 2.1. Multi-Dataset Object Detection

Multi-dataset object detection aims to bolster model generalization by synthesizing heterogeneous data sources. The primary challenge lies in reconciling label space inconsistency and domain-specific distribution shifts.Early approaches (Zhao et al., 2021; Yan et al., 2020) relied on manual taxonomy engineering via external knowledge graphs or semantic priors. However, these methods scale poorly as the number of categories grows. To address this, automated label fusion methods emerged; for instance, UniDet (Zhou et al., 2022) utilizes a prior-free taxonomy for scalable integration, while Universal-RCNN (Xu et al., 2020) employs task-specific attention to mitigate inter-dataset variance. The advent of Vision-Language Models (VLMs) has redefined label alignment. Recent works like ScaleDet (Chen et al., 2023b) leverage text embeddings to compute cross-dataset label similarities. Beyond semantic alignment, modern approaches focus on architectural scalability and training dynamics. For example, Plain-Det (Shi et al., 2024) demonstrates that minimalist architectures can excel in multi-dataset scenarios by reducing task-specific bias.

## 2.2. Visible-Infrared Object Detection

Benefiting from advances in infrared sensor technology, the acquisition of RGB-IR target detection data has become feasible, effectively overcoming the failure of visible models in complex environments such as low illumination, fog and occlusion. This technological progress has spurred the construction of drone-based dual-modal datasets like DroneVehicle (Sun et al., 2022) (for vehicle detection) and RGBTDronePerson (Zhang et al., 2023b) (for pedestrian detection), and corresponding dataset-specific vehicle (Liu et al., 2016; Yuan & Wei, 2024; Yuan et al., 2022) and pedestrian (Zhang et al., 2023b) detection algorithms, optimizing heterogeneous information fusion in aerial dual-modal object detection. However, existing algorithms lack a unified framework capable of simultaneously handling both vehicle and pedestrian targets. Inspired by these gaps, this study aims to build a drone-based dual-modal cross-task detection framework, jointly utilizing the DroneVehicle and RGBT-DronePerson datasets to achieve efficient parallel detection of vehicles and pedestrians in complex environments.

## 2.3. Mixture of Experts(MoE)

Mixture-of-Experts (MoE) has emerged as a fundamental paradigm for scaling neural network capacity while maintaining computational efficiency through sparse activation and dynamic routing (Jacobs et al., 1991). Following the success of V-MoE (Riquelme et al., 2021) in establishing this architecture within computer vision, the paradigm has rapidly evolved to support sophisticated multimodal and multi-task frameworks (Chen et al., 2023a; Sun et al., 2025). In the realm of object detection, dynamic expert allocation has proven particularly effective for processing heterogeneous data distributions. Notable instances include DAMEX (Jain et al., 2023), which introduces a dataset-aware mechanism to balance domain-specific features, and remote sensing architectures like SM3Det (Li et al., 2024) and SkySenseV2 (Zhang et al., 2025b), which leverage sparse MoEs to capture multi-scale features without prohibitive computational costs. Building upon these insights, this work develops a specialized MoE architecture tailored for drone-based unified detection, specifically addressing the unique task distribution shifts encountered in aerial scenarios.

## 3. Method

The overall architecture of DroneDINO, illustrated in Fig. 2, establishes a unified paradigm designed to flexibly process heterogeneous multi-modal and multi-dataset inputs through a shared DINO-based transformer stream. The processing pipeline commences with the HR-MoE module, which adaptively decouples universal representations, modality-related knowledge, and dynamic-aware features to address data heterogeneity. Subsequently, these refined em-

beddings traverse a weight-shared backbone and encoder; here, a Task-Recognition head is integrated to explicitly identify the input task category, thereby penalizing representations inconsistent with the identified task via a classification loss. Finally, the task-aware features are processed by a shared decoder and dataset-specific detection heads. Notably, by leveraging the common label space within individual datasets, DroneDINO maintains only three specialized detection heads—specifically for VisDrone, DroneVehicle, and RGBTDronePerson—to ensure optimal architectural scalability and computational efficiency.

### 3.1. Task Definition of Drone-UOD

We introduce Drone-based Unified Object Detection (Drone-UOD), a novel task requiring a single model to simultaneously generalize across heterogeneous datasets and adaptively handle three input modalities: visible (RGB), infrared (IR), and RGB-IR fusion. Building upon three representative drone-based datasets—VisDrone (RGB), DroneVehicle (RGB-IR), and RGBTDronePerson (RGB-IR)—we define five interrelated subtasks (Table 1) categorized by their input configurations, where Task 1 processes single RGB images $I_{RGB} \in \mathbb{R}^{H \times W \times 3}$, Tasks 2 and 4 process single IR images $I_{IR} \in \mathbb{R}^{H \times W \times 3}$, and Tasks 3 and 5 handle aligned image pairs $(I_{RGB}, I_{IR})$ for multi-modal fusion.

*Table 1.* Comparisons of subtasks from the Drone-UOD task.

| ID | MODALITY | TARGETS | DATA SOURCE |
|---|---|---|---|
| TASK 1 | RGB | VEHICLES + PEDESTRIANS | VISDRONE |
| TASK 2 | IR | VEHICLES | DRONEVEHICLE |
| TASK 3 | RGB-IR | VEHICLES | DRONEVEHICLE |
| TASK 4 | IR | PEDESTRIANS | RGBTDRONEPERSON |
| TASK 5 | RGB-IR | PEDESTRIANS | RGBTDRONEPERSON |

### 3.2. Unified Interface and Pipeline of DroneDINO

To enable DroneDINO to handle all subtasks within a single model, we first convert the heterogeneous inputs into a unified token embedding form $\mathbf{X} \in \mathbb{R}^{\frac{H}{P} \times \frac{W}{P} \times d}$, where $P$ is the patch size and $d$ is the embedding dimension. For single-modal inputs, modality-specific Patch Embedding functions $\text{PE}_{\text{RGB}}$ and $\text{PE}_{\text{IR}}$ are applied to generate tokens. For dual-modal inputs, an average fusion strategy is adopted to integrate the modalities:

$$\mathbf{X} = \begin{cases} \text{PE}_{\text{RGB}}(I_{RGB}) & ID = 1 \\ \text{PE}_{\text{IR}}(I_{IR}) & ID = 2, 4 \\ \frac{1}{2}(\text{PE}_{\text{RGB}}(I_{RGB}) + \text{PE}_{\text{IR}}(I_{IR})) & ID = 3, 5 \end{cases} \quad (1)$$

The unified tokens $\mathbf{X}$ are subsequently processed by the HR-MoE module to generate modulated embeddings, which is formulated as:

$$\mathbf{X}_{moe} = \text{HR-MoE}(\mathbf{X} \mid \mathbf{ID}). \quad (2)$$

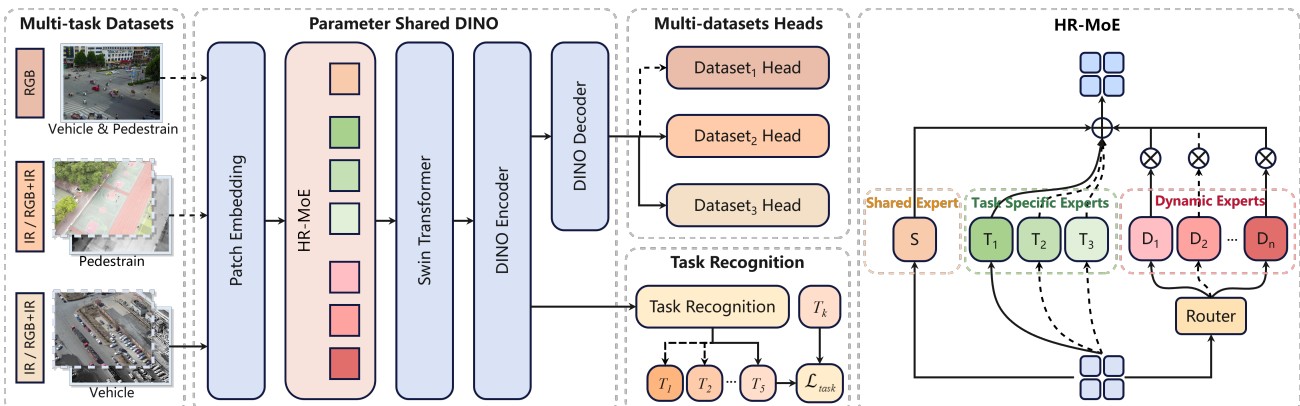

*Figure 2.* **Overall architecture of the DroneDINO framework.** We propose a unified paradigm to flexibly process multi-modal and multi-dataset inputs. The **left panel** illustrates the complete pipeline where input data flows into a shared DINO-based Transformer stream; here, the **Task-Recognition head** identifies task categories to penalize erroneous feature representations. The **right panel** details the internal structure of the **HR-MoE** module, which categorizes experts into Shared, Task-Specific, and Dynamic Expert group. This design aims to address modality heterogeneity and expert dominance in multi-task learning.

These modulated features are then fed into a shared Transformer stream, consisting of a Swin-Transformer backbone and a DINO encoder, to produce the task-aware encoded features:

$$\mathbf{X}_{enc} = \text{Encoder}(\text{Swin}(\mathbf{X}_{moe})). \qquad (3)$$

To supervise the feature learning process, the encoded features $\mathbf{X}_{enc}$ are passed into a Task-Recognition head $\psi$ to compute a task-specific classification where $\mathbf{ID}_{gt}$ denotes the ground-truth task identity:

$$\mathcal{L}_{task} = \mathcal{L}_{ce}(\psi(\mathbf{X}_{enc}), \mathbf{ID}_{gt}). \qquad (4)$$

Simultaneously, $\mathbf{X}_{enc}$ and a set of object queries $\mathbf{Q}$ are processed by a DINO decoder to generate decoder queries. Finally, these queries are fed into dataset-specific detection heads $\phi^{(ID)}$ to produce the final detection results:

$$\mathcal{B}, \mathcal{S}, \mathcal{C} = \phi^{(ID)}(\text{Decoder}(\mathbf{Q}, \mathbf{X}_{enc})) \qquad (5)$$

where $\mathcal{B}, \mathcal{S}$, and $\mathcal{C}$ represent the bounding box coordinates, confidence scores, and category labels.

### 3.3. Heterogeneous Routed Mixture of Experts

To effectively handle multi-task shifts while maintaining a unified representation, we propose the **HR-MoE** module. This module structurally decouples the feature space into universal invariant representations, task-specific features, and dynamic refinements through a tripartite forward flow involving a Shared Expert ($\Phi_{share}$), a task-specific expert group ($\Phi_{spec}$), and a dynamic expert group ($\Phi_{dyn}$). Formally, given the input $\mathbf{X} \in \mathbb{R}^{N \times C}$, the aggregated features $\tilde{\mathbf{Y}}$ and

the final residual output $\mathbf{Y}$ are formulated as:

$$\tilde{\mathbf{Y}} = \Phi_{share}(\mathbf{X}) + \Phi_{spec}(\mathbf{X} \mid \text{ID}) + \Phi_{dyn}(\mathbf{X}), \qquad (6)$$

$$\mathbf{Y} = \text{LN}_{final}\left(\tilde{\mathbf{Y}}\right) + \mathbf{X} \qquad (7)$$

where $\text{LN}_{final}$ denotes the final layer normalization, and ID is the task indicator.

**Shared and Task-Specific Experts.** These two stationary pathways are designed to capture different levels of data characteristics, addressing both commonality and discrepancy across tasks. The Shared Expert is incorporated to extract task-invariant visual patterns and stabilize the optimization landscape by preventing expert collapse. Simultaneously, the Task-Specific Experts are employed to explicitly align heterogeneous distributions and address the severe distributional shifts between different drone-based datasets. The forward processes for these pathways are defined as:

$$\Phi_{share}(\mathbf{X}) = \mathbf{W}_{share}^{proj}\left(\text{LN}_{share}\left(E_{share}(\mathbf{X})\right)\right), \qquad (8)$$

$$\Phi_{spec}(\mathbf{X} \mid ID) = \mathbf{W}_{ID}^{proj}\left(\text{LN}_{ID}\left(E_{ID}(\mathbf{X})\right)\right) \qquad (9)$$

where $E$ denote the expert network.

**Dynamic Experts and Routing.** The Dynamic Expert group handles intra-task variability through a token-level routing mechanism. This allows the model to specialize sub-modules for different visual patterns within the same task. The router computes noisy logits $h_{i,j}$ and sparse gating weights $G_{i,j}$ for each token $\mathbf{x}_i$:

$$h_{i,j} = (\mathbf{x}_i \cdot \mathbf{W}_g)_j + \epsilon_{i,j} \cdot \text{Softplus}((\mathbf{x}_i \cdot \mathbf{W}_n)_j), \qquad (10)$$

$$G_{i,j} = \frac{\exp(h_{i,j})}{\sum_{m \in \mathcal{T}_i} \exp(h_{i,m})} \cdot \mathbb{1}(j \in \mathcal{T}_i) \quad (11)$$

where $\epsilon \sim \mathcal{N}(0,1)$ and $\mathcal{T}_i$ denotes the set of Top-$k$ selected experts. To ensure stability in high-variance regimes, the dynamic expert output $\Phi_{\text{dyn}}$ is calculated using a Logarithmic Ensemble strategy to aggregate the outputs of selected experts:

$$\Phi_{\text{dyn}}(\mathbf{x}_i) = \ln\left(\sum_{j \in \mathcal{T}_i} G_{i,j} \cdot \exp(E_j(\mathbf{x}_i)) + \delta\right), \quad (12)$$

where $\delta$ is a stability constant. To prevent a few experts from dominating the routing process—which would reduce capacity and encourage collapse—we introduce a load balancing loss $\mathcal{L}_{\text{moe}}$. This loss monitors both the Usage Intensity $\mathbf{\Omega}$ (the total gating probability assigned to each expert) and the Selection Frequency $\mathbf{C}$ (the count of tokens assigned to each expert) across the batch:

$$\mathcal{L}_{\text{moe}} = \lambda_1 \underbrace{\frac{\text{Var}(\mathbf{\Omega})}{(\mathbb{E}[\mathbf{\Omega}])^2 + \epsilon}}_{\text{Weight Balance}} + \lambda_2 \underbrace{\frac{\text{Var}(\mathbf{C})}{(\mathbb{E}[\mathbf{C}])^2 + \epsilon}}_{\text{Count Balance}}. \quad (13)$$

By minimizing the squared coefficient of variation for both metrics, the model ensures an equitable distribution of both probability mass and computational load among all dynamic experts.

### 3.4. Task-Recognition Strategy

To ensure that input tokens $\mathbf{X}$ are adaptively routed to appropriate experts by HR-MoE and generate task-specific discriminative features by the Swin-Transformer backbone and DINO Encoder, we introduce a task-recognition strategy which is exclusively used during training. This strategy enables DroneDINO to differentiate between five tasks. We compute the average feature embedding $\mathbf{X}_{avg} \in R^{1 \times d}$ of the encoded features $\mathbf{X}_{enc} \in R^{N \times d}$:

$$\mathbf{X}_{avg} = \sum_{i=1}^{N} \mathbf{X}_{enc}^{(i)}, \quad (14)$$

where $N$ denotes the token number and $\mathbf{X}_{enc}^{(i)}$ represents the $i$-th token embedding. $\mathbf{X}_{avg}$ is then fed into a three-layer perceptron $\text{MLP}(\cdot)$ and classified into five tasks:

$$\mathbf{ID}_{pre} = \text{MLP}(\mathbf{X}_{avg}), \quad (15)$$

where $\mathbf{ID}_{pre} \in R^5$ represents the predicted probabilities for five tasks. The task-recognition loss is computed from $\mathbf{ID}_{pre}$ and its ground truth $\mathbf{ID}_{gt}$:

$$\mathcal{L}_{\text{task}} = \mathcal{L}_{\text{ce}}(\mathbf{ID}_{pre}, \mathbf{ID}_{gt}), \quad (16)$$

where $\mathcal{L}_{\text{ce}}$ denotes the cross-entropy loss.

### 3.5. Training Strategy and Loss Function

During training, we employ task-specialized batch sampling: for each iteration, we randomly select a task $ID \in \{1,2,3,4,5\}$ and construct the batch exclusively from the corresponding dataset. This ensures systematic coverage of all tasks while preventing optimization bias toward any single task. The architecture extends DINO with task-specific detection heads $\phi^{(ID)}$.

Parameter updates follow a task-conditional paradigm. When processing a batch from task $ID$, we optimize all parameters in DroneDINO except detection heads for the other four tasks. This selective update strategy eliminates interference between task-specific heads while maintaining gradient flow through shared components including the Swin-Transformer backbone, DINO encoder and decoder, and task-recognition MLP.

For all detection heads, following DINO, we use a weighted combination of the focal loss $\mathcal{L}_{\text{Focal}}$ for classification, L1 loss $\mathcal{L}_{\text{L1}}$ and generalized IoU loss $\mathcal{L}_{\text{GIoU}}$ for bounding box regression. The overall loss function is summarized as:

$$\begin{aligned} \mathcal{L}_{\text{det}} &= \mathcal{L}_{\text{Focal}} + \lambda_{\text{L1}}\mathcal{L}_{\text{L1}} + \lambda_{\text{GIoU}}\mathcal{L}_{\text{GIoU}}, \\ \mathcal{L}_{\text{total}} &= \mathcal{L}_{\text{det}} + \lambda_{\text{moe}}\mathcal{L}_{\text{moe}} + \lambda_{\text{task}}\mathcal{L}_{\text{task}} \end{aligned} \quad (17)$$

where $\lambda_{\text{L1}}, \lambda_{\text{GIoU}}, \lambda_{\text{moe}}, \lambda_{\text{task}}$ are regularization parameters.

During inference, we remove the task-recognition head to avoid unnecessary overhead and perform task-specific evaluation: for task $ID$, predictions are generated exclusively through $\phi^{(ID)}$ with MoE operating in deterministic mode (noise disabled and $\mathcal{L}_{\text{balance}}$ excluded).

## 4. Experiments

### 4.1. Dataset Setting

To validate the effectiveness of our proposed framework, we integrate three representative benchmarks—VisDrone, DroneVehicle, and RGBTDronePerson—which collectively establish the five sub-tasks described previously. VisDrone serves as a comprehensive baseline for RGB detection, containing 6,471 training and 548 testing images covering ten categories of vehicles and pedestrians. Regarding DroneVehicle, we converted its original oriented annotations into horizontal bounding boxes (denoted as DroneVehicle-HBB) to maintain annotation consistency across tasks. This dataset features 17,990 training and 8,980 testing aligned image pairs supporting IR, and RGB-IR vehicle detection. Complementing this, RGBTDronePerson focuses on pedestrian detection under varying illumination conditions, contributing 6,125 RGB-IR image pairs. All annotations are unified into a standardized horizontal bounding box format to facilitate joint training within the DroneDINO framework.

*Table 2.* Performance comparison with unified SOTA methods. The best results are highlighted in **red** and the second-best results are in **blue**.

| Model | Flops(G) | param(M) | VisDrone | | DroneVehicle | | | | RGBTDronePerson | |
|---|---|---|---|---|---|---|---|---|---|---|
| | | | Task 1 | | Task 2 | | Task 3 | | Task 4 | Task 5 |
| | | | $mAP$ | $mAP_{50}$ | $mAP$ | $mAP_{50}$ | $mAP$ | $mAP_{50}$ | $mAP_{50}$ | $AP_{50}$ |
| UniDet(Zhou et al., 2022) | 403 | 66 | 17.0 | 30.0 | 49.4 | 70.1 | - | - | 27.0 | - |
| CerberusDet(Tolstykh et al., 2025) | 505 | 143 | 23.3 | 39.2 | 60.5 | 81.5 | - | - | 40.4 | - |
| PlainDet(Shi et al., 2024) | 213 | 44 | 18.0 | 33.4 | 51.8 | 73.7 | - | - | 36.1 | - |
| SM3Det(Li et al., 2024) | 487 | 178 | 16.4 | 28.7 | 52.1 | 73.8 | - | - | 24.0 | - |
| Ours | 656 | 225 | **30.2** | **50.4** | **63.2** | **85.5** | **64.5** | **86.4** | **55.8** | **52.2** |

## 4.2. Implementation Details

All experiments were conducted on a computational cluster equipped with 4 NVIDIA GeForce RTX 4090 GPUs. The framework was implemented using PyTorch 2.1.2 with Python 3.11.11 and CUDA 11.8, built upon the MMDetection codebase. To ensure fair comparison across all models and ablation studies, we trained each configuration for 12 epochs using the AdamW optimizer with a base learning rate of $1.0 \times 10^{-3}$ and weight decay of $1.0 \times 10^{-4}$. The loss weighting scheme was configured with $\lambda_{cls} = 1$, $\lambda_{bbox} = 5$, $\lambda_{IoU} = 2$, $\lambda_{MoE} = 1$ and $\lambda_{task} = 1$. During training, we employed comprehensive data augmentation techniques including multi-scale random resizing, random horizontal/vertical flipping, and random cropping to enhance model robustness.

## 4.3. Comparison with State-of-the-Art Unified Detectors

To evaluate the effectiveness of DroneDINO, we benchmark it against leading unified object detection methods, including UniDet(Zhou et al., 2022), CerberusDet(Tolstykh et al., 2025), PlainDet(Shi et al., 2024), and SM3Det(Li et al., 2024). As shown in Table 2, while prior methods are restricted to single-modality inputs, our framework is designed to address the Drone-UOD task, successfully unifying all five sub-tasks within a single model. This design unlocks the potential of multi-modal fusion, yielding 64.5% mAP on Task 3 and 52.2% $mAP_{50}$ on Task 5.

Beyond its unique multi-modal capabilities, DroneDINO establishes a new state-of-the-art across the benchmarks, surpassing the runner-up, CerberusDet, by a substantial margin of 6.9% $mAP$ on VisDrone and 15.4% $mAP_{50}$ on RGBT-DronePerson. In addition, our model achieves 63.2% $mAP$ on the DroneVehicle IR detection task (Task 2). These quantitative gains are primarily attributed to the HR-MoE design, which resolves the expert conflict inherent in conventional sparse architectures by structurally decoupling parameter spaces into shared experts for universal visual patterns and task-specific experts for distinct feature distributions. Our approach effectively eliminates negative transfer across varied drone-based tasks, ensuring optimal performance for each individual source task within a single framework.

Figure 3 illustrates the comprehensive performance profile on the VisDrone dataset, where DroneDINO establishes a strictly dominant envelope, consistently outperforming state-of-the-art unified detectors across all semantic categories. While baselines like CerberusDet exhibit erratic inter-class variance, our approach maintains a robust high-performance equilibrium. This superiority validates that our architecture effectively resolves the optimization interference prevalent in multi-class learning. By decoupling category-specific representations through specialized expert routing, DroneDINO ensures that gradient updates for head categories do not suppress the feature learning of tail classes, achieving generalized mastery over diverse aerial targets.

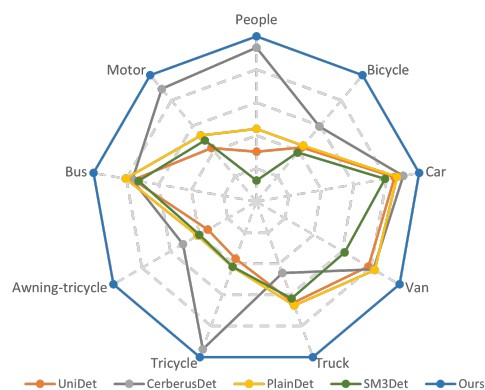

*Figure 3.* **Per-category performance on VisDrone.** The radar chart compares DroneDINO (blue) against state-of-the-art unified detectors. Note that DroneDINO forms the outermost envelope, demonstrating strictly superior and balanced performance across all categories compared to the high variance observed in baselines.

## 4.4. Comparison with State-of-the-Art Task-Specific Detectors

To demonstrate the robustness of our unified framework, we benchmark DroneDINO against specialized detectors tailored for each sub-task. As detailed in Tables 3, 4, and 5, our model consistently outperforms task-specific counterparts across diverse modalities.

**Results on VisDrone dataset:** We evaluate DroneDINO on the VisDrone dataset for RGB detection, comparing it against specialized state-of-the-art detectors including YOLO variants (Wang et al., 2024b;a; Khanam & Hus-

sain, 2024), QueryDet (Yang et al., 2022), HIC-YOLOv5 (Tang et al., 2024), and transformer-based models (Carion et al., 2020; Zhu et al., 2020; Roh et al., 2021; Zhao et al., 2024; Zhang et al., 2025a). Despite lacking task-specific customization, DroneDINO achieves 30.2% AP, surpassing highly optimized detectors including all YOLO variants (e.g., +3.9% AP over YOLOv10) and the domain-specialized QueryDet (+1.9% AP). While marginally trailed by UAV-DETR, our model demonstrates that a unified architecture can rival specialized designs in dense aerial scenarios.

*Table 3.* Performance comparison on VisDrone dataset. **Red** indicates the best performance, and blue indicates the second best.

| CATEGORY | METHOD | AP | AP$_{50}$ |
|---|---|---|---|
| YOLO SERIES | YOLOv8 | 26.1 | 42.7 |
| | YOLOv9 | 25.2 | 42.0 |
| | YOLOv10 | 26.3 | 43.1 |
| | YOLOv11 | 25.9 | 43.1 |
| DRONE SPECIFIC | QUERYDET | 28.3 | 48.1 |
| | HIC-YOLOv5 | 26.0 | 44.3 |
| TRANSFORMER-BASED | DETR | 24.1 | 40.1 |
| | DEFORMABLE-DETR | 27.1 | 42.2 |
| | SPARSE-DETR | 27.3 | 42.5 |
| | RT-DETR | 28.4 | 47.0 |
| | UAV-DETR | **31.5** | **51.1** |
| UNIFIED | DRONEDINO (OURS) | 30.2 | 50.4 |

**Results on DroneVehicle dataset:** DroneDINO exhibits superior generalization across both IR and RGB-IR tasks, outperforming representative detectors such as Faster R-CNN (Ren et al., 2015), DDQ-DETR (Zhang et al., 2023a), and RTMDet (Lyu et al., 2022).Under IR inputs, it achieves 63.2% mAP, outperforming the strong single-stage baseline RTMDet by +2.6%. Notably, in the RGB-IR fusion setting, our model attains 64.5% mAP, leading the best baseline by +11.3% (vs. RTMDet). The significant gains in high-precision metrics (e.g., mAP$_{75}$) further validate that our HR-MoE effectively leverages complementary multi-modal features to enhance localization accuracy.

*Table 4.* Performance comparison on DroneVehicle-HBB dataset. **Red** denotes the best result and blue denotes the second best.

| MODAL | METHOD | AP | AP$_{50}$ | AP$_{75}$ |
|---|---|---|---|---|
| IR | FASTER-RCNN | 48.1 | 71.1 | 57.5 |
| | DDQ-DETR | 56.9 | 78.9 | 69.4 |
| | RTMDET | 60.6 | 82.8 | 72.6 |
| | OURS | **63.2** | **85.5** | **75.6** |
| RGB-IR | FASTER-RCNN | 56.5 | 78.5 | 68.1 |
| | DDQ-DETR | 60.7 | 82.4 | 72.8 |
| | RTMDET | 53.2 | 77.1 | 64.5 |
| | OURS | **64.5** | **86.4** | **77.6** |

**Results on RGBTDronePerson dataset:** On this challeng-

ing fine-grained benchmark, DroneDINO establishes new state-of-the-art performance for both IR and RGB-IR pedestrian detection. Compared against specialized IR detectors (Ren et al., 2015; Lyu et al., 2022; Zhang et al., 2023a) and advanced fusion methods including HRFuser (Broedermann et al., 2023), TINet (Zhang et al., 2023c), ICAFusion (Shen et al., 2024), C2Former (Yuan & Wei, 2024), QFDet (Zhang et al., 2023b), and PPSDet (Zhang et al., 2025c), our unified model achieves 55.8% AP$_{50}$ for IR detection, surpassing DDQ-DETR by +2.3% . In the complex RGB-IR fusion scenario, it delivers 52.2% AP$_{50}$, outperforming the previous SOTA (PPSDet) by +8.6%. These results confirm that DroneDINO successfully mitigates the negative transfer often seen in unified learning, delivering best-in-class performance even on tail categories like Crowd and Rider.

*Table 5.* Performance comparison on RGBTDronePerson dataset. **Red** denotes the best result and blue denotes the second best.

| MODAL | METHOD | Person | Rider | Crowd | AP$_{50}$ |
|---|---|---|---|---|---|
| IR | FASTER-RCNN | 33.8 | 39.5 | 32.3 | 35.2 |
| | RTMDET | 34.6 | 30.2 | 28.5 | 31.1 |
| | DDQ-DETR | 60.6 | 55.7 | **44.2** | 53.5 |
| | OURS | **64.0** | **59.8** | 43.7 | **55.8** |
| RGB-IR | HRFUSER | 16.3 | 24.8 | 25.6 | 22.2 |
| | TINET | 15.2 | 43.4 | 26.3 | 28.3 |
| | ICAFUSION | 28.6 | 19.3 | 28.0 | 28.6 |
| | C2FORMER | 37.4 | 45.6 | 30.2 | 37.7 |
| | QFDET | 46.1 | 50.3 | 29.9 | 42.1 |
| | PPSDET | 49.8 | 52.6 | 28.3 | 43.6 |
| | OURS | **61.2** | **55.0** | **40.4** | **52.2** |

### 4.5. Ablation and Analysis

**Effectiveness of Key Components.** We analyze the contribution of each module in Table. 6.

*Table 6.* Ablation study on different components. The numbers denote the performance change (AP) relative to our full model.The notation "-" indicates the removal of a specific component.

| METHOD | $T_1$ | $T_2$ | $T_3$ | $T_4$ | $T_5$ | $\Delta$AP |
|---|---|---|---|---|---|---|
| OURS | – | – | – | – | – | – |
| −TASK-RECOGNITION HEAD | +0.6 | +0.1 | -0.2 | -0.6 | -1.7 | -0.36 |
| HR-MoE → MoE | -2.2 | -0.3 | -0.1 | -1.8 | -3.4 | -1.56 |
| −HR-MoE | -3.1 | -1.0 | -0.9 | -2.5 | -3.0 | -2.10 |
| −MULTI-TASK | -1.8 | -1.6 | -1.3 | -18.1 | -15.8 | -7.72 |

First, the HR-MoE design is pivotal; replacing it with a standard MoE degrades performance by 1.56 AP. This confirms that homogeneous routing suffers from expert collapse, where experts gravitate toward dominant tasks (e.g., $T_1$) due to gradient magnitude disparities. In contrast, our heterogeneous grouping explicitly decouples shared and task-specific representations, preventing such interference.

Second, the Task Recognition head serves as a critical semantic regularizer. Its removal significantly harms fusion tasks ($T_3, T_5$), indicating that explicit task conditioning is required to align multi-modal feature manifolds and prevent the encoder from overfitting to the statistical modes of a

single modality.

Finally, the Unified Training strategy proves essential for data-scarce tasks ($T_4, T_5$), facilitating positive knowledge transfer from data-rich domains to prevent convergence failure.

**Qualitative Visualization.** To intuitively verify the impact of HR-MoE on feature representational power, we visualize the activation maps in Figure 4. The baseline detector, hampered by sub-optimal feature coupling, exhibits spatially diffuse activations that frequently spill over into background clutter and fail to delineate adjacent small targets in dense scenarios. In sharp contrast, DroneDINO equipped with HR-MoE generates highly concentrated response maps that align strictly with object boundaries. This distinctive improvement demonstrates that our heterogeneous routing mechanism effectively mitigates feature blurring by assigning distinct semantic patterns to specialized experts. Consequently, the model successfully suppresses task-irrelevant environmental noise while enhancing instance-level discriminability, validating that the HR-MoE design is pivotal for extracting robust, object-centric features in complex aerial imagery.

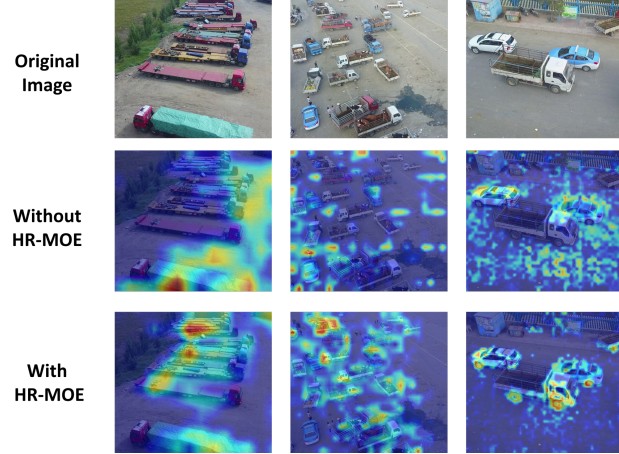

*Figure 4.* Visualization of feature heatmaps generated by superimposing aggregated backbone feature maps onto original images. The middle row displays the baseline model results, while the bottom row highlights our DroneDINO with HR-MoE.

**Expert Configuration Analysis.** To determine the optimal trade-off between model capacity and training stability, we conduct a comprehensive ablation study on expert scalability, as detailed in Table 7. Initially, by varying the total expert pool, we observe that increasing the count from 4 to 8 enhances performance due to the expanded representational capacity for diverse targets; however, further scaling to 16 results in diminishing returns (reverting to 63.3% mAP), which we attribute to gradient sparsity preventing excessive experts from receiving sufficient optimization signals.

Subsequently, with the total pool fixed at 8, we analyze the routing strategy: the Top-4 activation yields the peak performance of 63.9% mAP, significantly outperforming the fully dense execution (Top-8, 63.0% mAP). This indicates that the Top-4 setting strikes a critical balance, fostering sufficient expert collaboration to capture complex multi-modal variances while retaining the sparsity necessary for expert specialization, thus serving as our default configuration.

*Table 7.* Ablation study on expert configurations for DroneVehicle detection.

| CONFIGURATION | AP | AP$_{50}$ | AP$_{75}$ |
|---|---|---|---|
| **TOTAL EXPERTS VARIATION (ACTIVATED = TOTAL/4)** | | | |
| 4 EXPERTS (ACT: 1) | 63.3 | 85.4 | 76.2 |
| 8 EXPERTS (ACT: 2) | 63.6 | 85.9 | 76.5 |
| 16 EXPERTS (ACT: 4) | 63.3 | 85.5 | 76.0 |
| **ACTIVATED EXPERTS VARIATION (TOTAL = 8)** | | | |
| 2 ACTIVATED | 63.6 | 85.9 | 76.5 |
| 4 ACTIVATED | **63.9** | **86.3** | **76.6** |
| 8 ACTIVATED | 63.0 | 85.1 | 75.8 |

## 5. Conclusion and Limitation

In this paper, we address modality heterogeneity and architectural scalability in drone perception by proposing Drone-UOD, a unified task requiring simultaneous handling of RGB, IR, and multi-modal fusion. We introduce DroneDINO, a framework featuring HR-MoE and a task-recognition auxiliary strategy. By decoupling experts into shared, task-specific, and dynamic groups, HR-MoE resolves expert conflict and ensures balanced feature learning across heterogeneous tasks. Meanwhile, the task-recognition auxiliary loss constrains the feature space by penalizing representations inconsistent with the target task. Extensive experiments on three benchmarks validate that DroneDINO unifies five sub-tasks within a single model and outperforms state-of-the-art specialized detectors. This work provides a robust baseline for building all-rounder foundation models in low-altitude aerial applications.

While significant advancements, several limitations remain that define promising avenues for future research. DroneDINO is restricted to horizontal detection, suggesting oriented bounding boxes as a necessary extension for aerial robustness. Additionally, transitioning from multiple task-specific heads to a unified interface represents a key scalability goal to handle open-ended label spaces. Finally, incorporating diverse modalities like SAR or hyperspectral data and expanding into tasks such as segmentation will be crucial for evolving this framework into a universal foundation model for autonomous drone perception.

## Impact Statement

This paper presents work whose goal is to advance the field of Machine Learning. There are many potential societal consequences of our work, none which we feel must be specifically highlighted here.

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
