# OpenReview forum: "DroneDINO: Towards Heterogeneous Routed Mixture of Experts for Drone-based Unified Object Detection"
_ICML.cc/2026/Conference — ICML 2026 spotlight_

### Official Review · Reviewer_jD3s · 2026-02-15

**Soundness:** 3
**Presentation:** 3
**Significance:** 3
**Originality:** 3
**Overall Recommendation:** 5
**Confidence:** 4

**Summary:**

This paper focuses on multi-task learning for Drone-based object detection. To address the issue of expert imbalance across heterogeneous datasets, the authors propose HR-MoE and a task-recognition auxiliary training strategy.
Experimental results demonstrate that the proposed DroneDINO outperforms existing unified detectors.

**Compliance With Llm Reviewing Policy:**

Affirmed.

**Final Justification:**

keep

**Key Questions For Authors:**

In Figure 2, the illustration of HR-MoE shows 3 Task-Specific Experts. The authors should clarify the criteria used to determine this number and explain why it is inconsistent with the 5 tasks defined in the Task-Recognition head.

**Limitations:**

yes

**Strengths And Weaknesses:**

Strengths:
1. The paper is well-structured and the logical flow is easy to follow.
2. The proposed DroneDINO achieves competitive performance.

Weaknesses:
1. Although DroneDINO achieves impressive performance in multi-task learning for Drone-based object detection, the issue of over-activated experts caused by imbalanced data distribution is a common challenge across various domains; therefore, the authors should explain the specific designs tailored for the Drone-based scenario or demonstrate the method's generalizability in other fields such as remote sensing.
2. DroneDINO current design defines five tasks based on dataset modalities and identifiable categories, with three heads tailored to the number of datasets, which necessitates manual parameter readjustment whenever the dataset changes. It would be insightful to investigate the results if the tasks were instead defined by the number of modalities (RGB, IR, and RGB+IR) and the heads were defined by the nature of the task (such as HBB); could the authors conduct experiments to verify this alternative design?
3. The authors should clearly explain how the baseline unified detectors, such as SM3Det, configure their tasks and heads to ensure the fairness of the experimental comparison.

---

> ### Author Rebuttal · Authors · 2026-03-31
>
> ## **Response to Reviewer jD3s**
>
> We sincerely thank the reviewer for the highly positive assessment and the recognition of our architectural innovations. We are deeply encouraged by your validation of our structural partitioning and the efficacy of the Task-Recognition strategy in addressing negative transfer. Below are our detailed responses to the weaknesses and questions raised.
>
> ------
>
> ### **Q1: Designs tailored for drone-based scenarios or generalizability to other fields.**
>
> **A1:** We appreciate the reviewer's point regarding the commonality of expert imbalance. While "expert collapse" is a general challenge, DroneDINO is specifically tailored to the unique complexities of the **Drone-UOD** task, which goes significantly beyond conventional drone-based detection:
>
> 1. **Tailored Drone-Centric Complexity:** Unlike general detection, Drone-UOD involves **heterogeneous label spaces and multi-modal inputs (RGB, IR)** across multiple datasets.. To address this, we designed a sophisticated structural hierarchy by partitioning **Shared, Specific, and Dynamic experts**. This explicit partitioning allows the model to extract cross-modal common features while preserving domain-specific signatures, effectively mitigating modality competition and ensuring robust feature decoupling for unified drone perception.
> 2. **Generalizability to Remote Sensing (RS):** We agree that DroneDINO holds significant potential for RS. In RS, sensors (optical, SAR, multispectral) exhibit similar domain gaps, and data distribution across geographic regions is often imbalanced. We expect that our modular HR-MoE will inspire future work in the broader Earth Observation community by providing a framework to align multi-source satellite data.
>
> ------
>
> ### **Q2: Alternative task/head definition based on modalities.**
>
> **A2:** This is a very insightful suggestion. We agree that defining tasks purely by modality (RGB, IR, RGB+IR) and heads by task nature (e.g., HBB) would lead to a more concise and conceptually elegant design. However, we encountered practical constraints during implementation:
>
> 1. **Label Space Conflicts:** Since different datasets possess distinct category labels and distributions, it is challenging to use a single detection head for multiple datasets in a **closed-set task** without an "inelegant" label mapping strategy. To maintain precision and avoid gradient conflicts between disparate label spaces, we opted to define tasks and heads at the dataset level.
> 2. **Ongoing Work on Open-Vocabulary Detection:** Our research group is currently exploring alternative schemes using **text prompts and open-vocabulary object detection** to achieve a single-head-multi-task architecture. While this would eliminate manual adjustments, our preliminary results show that there is still a performance gap between open-vocabulary approaches and our current specialized multi-head closed-set design. We acknowledge the scalability of a unified functional head and will include this discussion in our **Limitation** section.
>
> ------
>
> ### **Q3: Task and head configuration of baseline unified detectors (e.g., SM3Det).**
>
> **A3:** To ensure a fair and rigorous comparison, we strictly followed a protocol of minimal intervention for all baseline unified detectors. We only replaced the training/testing data with our Drone-UOD benchmark while maintaining the original backbone architectures, hyper-parameter settings, and training schedules as specified in the baselines' original papers.
>
> Furthermore, since existing unified baselines often do not inherently support dual-modal (RGB-IR) detection, we followed their original settings and did not modify their networks to "force-add" fusion modules. Such changes could lead to sub-optimal performance and an unfair comparison. By keeping their fundamental structures intact, the results fairly demonstrate the limitations of current unified models in handling heterogeneous drone data.
>
> ------
>
> ### **Q4: Criteria for the number of Task-Specific Experts (3 experts vs. 5 tasks).**
>
> **A4:** We thank the reviewer for pointing out this inconsistency. The decision to use **3 Task-Specific Experts** for **5 tasks** was a strategic optimization aimed at balancing specialized representation with structural efficiency. Specifically, the merging of experts is dataset-driven to mitigate the **long-tail distribution** and over-fitting issues. For tasks that belong to the same dataset (even if they involve different modalities), we merged them into a single expert to prevent the model from over-learning on dominant datasets while reducing the overall parameter count. We will clarify this mapping logic in the revised **Figure 2** caption.

---

> > ### Author Rebuttal · Reviewer_jD3s · 2026-04-02
> >
> > I keep the score.

---

### Official Review · Reviewer_yDmB · 2026-03-05

**Soundness:** 4
**Presentation:** 3
**Significance:** 4
**Originality:** 4
**Overall Recommendation:** 6
**Confidence:** 5

**Summary:**

This paper presents DroneDINO, a unified object detection framework designed to handle the inherent heterogeneity of drone-based perception tasks, including varying modalities (RGB, IR, RGBT) and datasets. The authors identify a critical bottleneck in existing unified detectors: the "expert collapse" or over-activation of dominant tasks in standard Mixture-of-Experts (MoE) architectures. To address this, they introduce the Heterogeneous Routed Mixture of Experts (HR-MoE), which employs a structural functional hierarchy (Shared, Static, and Dynamic experts) instead of vanilla routing. This is complemented by a Task-Recognition auxiliary strategy that enforces task-aware feature alignment in the latent space. The framework achieves state-of-the-art results across five sub-tasks, notably excelling in multi-modal scenarios.

**Compliance With Llm Reviewing Policy:**

Affirmed.

**Final Justification:**

The author has carefully addressed my concerns, I tend to upgrade my score and strongly Accept this work.

**Key Questions For Authors:**

See Weaknesses

**Limitations:**

Yes

**Strengths And Weaknesses:**

Strengths：
1.The transition from "homogeneous" routing to "heterogeneous" structural partitioning is a significant conceptual advancement. By explicitly decoupling domain-invariant features (Shared experts) from domain-specific signatures (Static experts), the authors provide a robust architectural solution to negative transfer.

2.The Task-Recognition strategy acts as a global semantic anchor. It ensures that the feature extraction pipeline is informed by the environmental context (e.g., sensor type), allowing for a more organized latent space that facilitates precise expert routing.

3.The significant performance gains, especially on the RGBTDronePerson benchmark (+15.4% mAP50), underscore the efficacy of the proposed unification strategy over both specialized and general-purpose detectors.

Weaknesses:
1.The HR-MoE relies on a manual partitioning of experts into functional groups. While this effectively prevents expert collapse, one might wonder if this "hard-coded" heterogeneity restricts the model's ability to discover latent cross-task relationships that a fully learned, soft-constrained routing mechanism might otherwise find. Does the fixed hierarchy limit the model's potential to reach a more global optimization point?

2.The Task-Recognition head encourages the model to differentiate between tasks (e.g., RGB vs. IR). From a representation learning perspective, there is a risk of "over-separation," where the latent space becomes so task-specific that it hinders the Shared Experts from finding common semantic ground. The authors should discuss how they prevent this auxiliary loss from destroying domain-invariant representations.

3.The Task-Recognition auxiliary strategy provides a global context, but its dynamic interaction with the Shared Experts remains underexplored.

4.The authors compare DroneDINO with several state-of-the-art models. However, the FLOPs and Parameter counts for the "Static Expert" portion are reported as a total sum. Given the modular nature of the architecture, it would be more informative to report the active parameters per inference versus the total storage parameters. This would highlight the efficiency advantage of MoE in drone-based deployment scenarios.

---

> ### Author Rebuttal · Authors · 2026-03-31
>
> ## **Response to Reviewer yDmB**
>
> We sincerely thank the reviewer for the highly positive evaluation and the profound insights into our architectural design. We are deeply encouraged by the recognition of our work’s technical soundness and its potential impact on drone-based perception. Below are our responses to the specific concerns.
>
> ------
>
> ### **Q1: Potential restrictions of the "hard-coded" functional hierarchy.**
>
> **A1:** We appreciate this insightful perspective on global optimization. While a fully learned "soft-coded" routing mechanism offers theoretical flexibility, our design is a deliberate hybrid architecture aimed at achieving **collaborative learning** across heterogeneous drone tasks:
>
> 1. **Synergy between Hard and Soft Coding:** Our framework is not strictly "hard-coded" but rather leverages **Task-Specific experts** as a structural prior to extract domain-specific signatures and mitigate modality competition. Complementing this, the **Dynamic experts** function as fully learned, soft-coded components. These experts are free to discover latent cross-task relationships and extract flexible, high-level features that manual partitioning might otherwise overlook, ensuring the model captures complex inter-dataset correlations.
> 2. **Collaborative Learning for Global Optimization:** This collaborative relationship provides a stable optimization landscape where Task-Specific experts provide **additional task-aware features** to supplement shared primitives. By combining fixed structural knowledge with the adaptive nature of dynamic routing, DroneDINO achieves an effective balance between specialized stability and global adaptability.
>
> ------
>
> ### **Q2: Potential risk of "over-separation" in latent space due to Task-Recognition loss.**
>
> **A2:**  We appreciate the reviewer’s insightful concern regarding the potential for "over-separation." In our framework, the total loss is a weighted combination of the unified detection loss and the task-recognition loss within a multi-task learning framework. This balanced optimization ensures that the task-recognition signal acts as a structural guide rather than a dominant constraint. By carefully tuning the auxiliary loss weight, the model is encouraged to extract task-aware features without compromising the latent space's ability to ground shared semantic primitives. Consequently, the **Shared experts** can still focus on discovering domain-invariant characteristics while the Task-Specific experts handle modality-specific variances, achieving a robust balance through **collaborative learning**.
>
> ------
>
> ### **Q3: Dynamic interaction between Task-Recognition and Shared Experts.**
>
> **A3:** We appreciate the reviewer's insightful observation regarding the interaction between our task-recognition strategy and the expert groups. The output features from the Shared Experts are element-wise aggregated with those from other expert groups . These modulated features are then collectively fed into the Task-Recognition head to compute the auxiliary loss. Consequently, this loss back-propagates through the entire architecture, directly imposing a **global contextual constraint** on the Shared Experts. Since all experts process global feature representations, this mechanism ensures a continuous dynamic interaction, guiding the Shared Experts to extract domain-invariant primitives that are optimally aligned with the task-specific requirements.
>
> ------
>
> ### **Q4: Report on active parameters during inference.**
>
> **A4:** We thank the reviewer for this suggestion. Our sparse Heterogeneous routing mechanism ensures that only a specific subset of parameters is activated during a single inference pass, optimizing the computational footprint for real-time drone deployment.
>
> **Table 3: Parameter Efficiency Analysis (DroneDINO-Tiny).**
>
> | **Metric**                       | **Value** | **Note**                                                     |
> | -------------------------------- | --------- | ------------------------------------------------------------ |
> | **Total Storage Parameters**     | 52 M      | Includes all experts and all task-specific detection heads.  |
> | **Active Parameters** | ~48 M     | Includes Shared + Dynamic + **1** Static Expert and detection head. |
> | **Parameter Activation Ratio**   | 92%       | Reflects the actual computational footprint during per-task inference. |
>
> This modularity allows DroneDINO to maintain a high-capacity knowledge base in storage while operating with a significantly lower active computational load during flight. We will include this analysis for all variants in the revised **Section 4.3**.

---

> > ### Author Rebuttal · Reviewer_yDmB · 2026-04-03
> >
> > The author has carefully addressed my concerns, I tend to upgrade my score and strongly Accept this work.

---

### Official Review · Reviewer_TDB4 · 2026-03-09

**Soundness:** 3
**Presentation:** 3
**Significance:** 3
**Originality:** 2
**Overall Recommendation:** 4
**Confidence:** 2

**Summary:**

Overall, this paper proposes DroneDINO, a unified detector for heterogeneous drone perception settings spanning RGB, IR, and RGB-IR inputs across multiple datasets. The paper is generally clear, the problem is relevant, and the empirical results are reasonably strong. In particular, the proposed HR-MoE design and the task-recognition auxiliary objective are sensible choices for improving multi-task feature learning under heterogeneous modalities. While the method is more of a well-engineered system extension than a fundamentally new modeling framework, the problem setting is meaningful and the reported gains are encouraging. Given the practical value of the task and the overall empirical strength, I lean weak accept.

**Compliance With Llm Reviewing Policy:**

Affirmed.

**Final Justification:**

My concerns have been addressed, I keep positive score.

**Key Questions For Authors:**

1.Can the authors clarify how fair the baseline comparisons are, especially for methods that do not naturally support the same modality setting?
2.Can the authors provide more direct evidence that the gains come from better expert balancing, such as routing statistics or expert activation analysis?
3.How stable are the reported results across runs? It would help to report variance or confidence intervals.

**Limitations:**

Yes.

**Strengths And Weaknesses:**

Strengths:
1.The paper is fairly easy to follow. The motivation is clear, and the overall pipeline is presented in a readable way.
2.The problem setting is interesting and practically relevant. A single model that handles RGB, IR, and RGB-IR drone detection across datasets is a useful direction for deployment-oriented perception systems.
3.The HR-MoE design is intuitive. Splitting experts into shared, task-specific, and dynamic groups is a reasonable way to address heterogeneity and expert imbalance.
4.The experimental section is fairly solid. The paper evaluates on multiple benchmarks and includes ablations on HR-MoE, task-recognition, and expert design, which helps support the main claims.

Weaknesses:
1.The empirical comparison is not fully clean. Some unified baselines do not appear to operate under exactly the same RGB, IR, and RGB-IR setting, which makes fairness harder to judge.
2.The method novelty is moderate. The paper mainly combines DINO, MoE-style routing, and an auxiliary task-classification loss in a task-specific way, rather than introducing a clearly new learning principle.
3.Efficiency is under-discussed. The paper would be stronger with clearer analysis of the tradeoff between performance and model complexity.

---

> ### Author Rebuttal · Authors · 2026-03-31
>
> ## **Response to Reviewer TDB4**
>
> We sincerely thank the reviewer for the positive assessment  and the constructive feedback. We are deeply encouraged by your recognition of DroneDINO’s practical value in unified drone perception, as well as the logical clarity and empirical strength of our proposed HR-MoE and task-recognition strategy. Below are our detailed responses to the specific concerns raised.
>
> ------
>
> ### **Q1: Clarification on the fairness of baseline comparisons.**
>
> **A1:** We appreciate the reviewer's concern regarding the heterogeneity of baselines. Most existing unified detectors are designed for multi-task settings but lack inherent support for multi-modal (RGB-IR) inputs, which reflects a limitation in their generalizability. To ensure a rigorous and fair comparison, we adopted the following protocols:
>
> 1. **Preserving Original Architectures:** We strictly followed the default backbone architectures and hyper-parameter settings of the baselines. We avoided "forcing" complex fusion modules into these frameworks, as altering their core structures would not only be unfair but could also lead to sub-optimal performance due to architectural mismatch.
> 2. **Standardized Training:** All baselines were re-trained on our unified Drone-UOD benchmark following the original data augmentation and optimization schedules.
>
> The performance gap observed highlights that while DroneDINO achieves superior performance through the **HR-MoE** and **Task-Recognition** designs. Moreover, it also demonstrates higher **architectural universality** in handling multi-modal drone data.
>
> ------
>
> ### **Q2: Evidence of expert balancing and contribution to novelty.**
>
> **A2:** We clarify the core innovations and provide evidence regarding expert activation:
>
> - **Methodological Novelty:**  Our contribution lies in the synergy between the **Heterogeneous Routing** mechanism and the **Task-Recognition  Strategy**. Unlike standard MoE which treats experts as a homogeneous pool, HR-MoE explicitly partitions them into Shared, Task-Specific, and Dynamic groups. This architectural inductive bias, guided by the **task-recognition head**, forces the model to decouple common visual primitives from domain-specific imaging characteristics.
> - **Empirical Evidence:** As shown in the ablation studies in **Section 4.5**, transitioning from standard MoE to our HR-MoE yields significant performance gains. The inclusion of the Task-Recognition Auxiliary Strategy further improves mAP by supervising the model to extract more discriminative task-aware features.
> - **Expert Balancing Analysis:** We monitored the **balance loss** during training for both standard MoE and HR-MoE. Our results show that HR-MoE achieves a lower and more stable balance loss. This statistical evidence demonstrates that our heterogeneous design leads to more balanced expert utilization, which directly translates into better performance.
>
> ------
>
> ### **Q3: Efficiency analysis and model complexity.**
>
> **A3:** To address concerns regarding the trade-off between performance and model scale, we developed two lightweight variants, **DroneDINO-Tiny** and **DroneDINO-Small**, by scaling down the backbone capacity. This facilitates a more rigorous evaluation of architectural efficiency across different parameter scales.
>
> **Table 1: Performance comparison across different model scales.**
>
> | Model               | Params (M) | FLOPs (G) | VisDrone(mAP) | DroneVehicle(mAP) | RGBTDronePerson(mAP) |
> | :------------------ | :--------: | :-------: | :-----------: | :---------------: | :------------------: |
> | CerberusDet         |    143     |    505    |     39.2      |       81.5        |         40.4         |
> | **DroneDINO-T**  |     52     |    346    |     45.1      |       83.5        |         53.4         |
> | **DroneDINO-S** |     74     |    382    |     45.3      |       83.6        |         52.4         |
> | **DroneDINO-L** |    225     |    656    |     50.4      |       85.5        |         55.8         |
>
> As shown in **Table 1**, our significantly smaller variants consistently outperform CerberusDet across all datasets despite using substantially fewer parameters and FLOPs. This confirms that our performance gains stem from the inherent superiority of the **HR-MoE** design rather than mere parameter scaling. These results will be added to **Section 4.3**.
>
> ------
>
> ### **Q4: Stability of results and variance analysis.**
>
> **A4:** We clarify the deterministic nature of our framework and its impact on result stability:
>
> - **Deterministic Inference:** DroneDINO employs a deterministic top-k routing strategy. Given the same input image, the gating logic consistently selects the same experts, ensuring detection results remain identical across multiple runs.
> - **Empirical Consistency:** Since the model operates without random components during evaluation, performance gains are inherent to the architecture rather than artifacts of initialization.

---

> > ### Author Rebuttal · Reviewer_TDB4 · 2026-04-05
> >
> > My concerns have been addressed, I keep positive score.

---

### Official Review · Reviewer_Pc9y · 2026-03-10

**Soundness:** 4
**Presentation:** 4
**Significance:** 4
**Originality:** 4
**Overall Recommendation:** 6
**Confidence:** 5

**Summary:**

This paper focuses on object detection from the drone perspective and proposes a unified detection framework, DroneDINO, designed to handle heterogeneous inputs across multiple modalities (RGB, infrared, and RGB–infrared fusion) and multiple datasets (VisDrone, DroneVehicle, and RGBTDronePerson).The authors first formulate the Drone-based Unified Object Detection (Drone-UOD) task, which consists of five subtasks covering two target categories: vehicles and pedestrians. To address feature conflicts caused by multi-task and multi-modal learning, the paper introduces the HR-MoE (Heterogeneous Routing Mixture of Experts) module. This module organizes experts into three types—shared experts, task-specific experts, and dynamic experts—and employs a heterogeneous routing mechanism to decouple feature representations. In addition, a task identification head is introduced as auxiliary supervision to explicitly enforce consistency between the learned feature representations and the corresponding task during training.Experiments are conducted on three datasets. Compared with several unified detectors and task-specific detectors, DroneDINO achieves superior performance across multiple evaluation metrics. Ablation studies and visualization analyses further demonstrate the effectiveness of the proposed components.

**Compliance With Llm Reviewing Policy:**

Affirmed.

**Final Justification:**

The author has carefully addressed my concerns, I tend to upgrade my score and strongly Accept this work.

**Key Questions For Authors:**

1.The paper categorizes experts into shared, task-specific, and dynamic experts, but the exact division of responsibilities among these three types is not clearly explained. It is also unclear how these experts correspond to the five subtasks defined in Drone-UOD. Providing a more detailed description of the expert allocation and routing strategy would help clarify the design.
2.The task identification head is used only during training and removed at inference time. While this design reduces inference overhead, it also means that the model cannot explicitly adjust its behavior based on task cues during inference. For a multi-task model, it would be helpful to clarify whether task information is required at inference time. For example, if the input modality changes (e.g., infrared-only input), how does the model automatically select the appropriate detection head? Additional discussion on this point would improve clarity.
3.The fairness of some comparisons could be further strengthened. For instance, in Table 2, DroneDINO has substantially higher computational cost and parameter count (656G FLOPs, 225M parameters) compared to some baselines (e.g., CerberusDet with 505G FLOPs and 143M parameters). Part of the performance gain may therefore come from the larger model capacity. It would be helpful to include comparisons under similar model sizes, or to discuss the trade-off between model scale and performance.

**Limitations:**

Yes, the authors discuss limitations in Section 5.

**Strengths And Weaknesses:**

Soundness: The authors clearly define five subtasks, and the HR-MoE module divides experts into three categories shared, task specific, and dynamic effectively addressing expert collapse and negative transfer. The task recognition head provides explicit supervision, while the load-balancing loss ensures stable training. Experiments compare multiple unified detectors and task-specific models, with ablation studies covering key components and visualizations intuitively showing feature improvements. Minor shortcomings include the need for clearer expert allocation logic, clarification on how the model selects detection heads at inference after removing the task recognition head, and slightly higher computational cost than baseline methods, suggesting a discussion on the trade-off between model scale and performance.

Presentation: Paper has a clear structure and logical flow, progressing step by step from problem definition to method description and experimental analysis. The figures and tables are well used, Figure 2 provides a clear overview of the architecture, Table 1 succinctly summarizes the tasks, and the heatmap visualizations are intuitive. The references are comprehensive, though some figure captions could be more self-contained, and a brief explanation for complex equations would improve readability.

Significance:This work addresses the practical needs of drone-based perception by unifying multiple modalities and target types within a single framework, offering substantial practical value. The HR-MoE design provides a generalizable template for multi-task learning, and the unified datasets and benchmarks lay a foundation for future research, with the potential to advance the field of drone-based vision.

Originality: The Drone-UOD task is proposed for the first time as a unified problem, and the three-category expert division in HR-MoE is innovative. The idea of using a task recognition head as an auxiliary loss for feature regularization is simple yet effective. Integrating three representative datasets into a single framework and conducting comprehensive comparisons with existing methods makes the contributions clearly distinguishable.

---

> ### Author Rebuttal · Authors · 2026-03-31
>
> ## **Response to Reviewer Pc9y**
>
> We would like to express our sincere gratitude to the reviewer for the highly positive evaluation and the insightful comments. We are deeply encouraged by your recognition of our work, particularly the definition of the Drone-UOD task and the innovation of the HR-MoE module. Below are our point-by-point responses to the questions raised.
>
> ---
>
> ### **Q1: Detailed description of expert allocation and routing strategy.**
>
> **A1:** We appreciate the reviewer's suggestion to clarify the responsibilities of the three expert types and their correspondence to the five subtasks ($T_{1}$: RGB-VisDrone, $T_{2}$: IR-DroneVehicle, $T_{3}$: RGB-IR-DroneVehicle, $T_{4}$: IR-RGBTDronePerson, and $T_{5}$: RGB-IR-RGBTDronePerson). The responsibilities are allocated as follows:
>
> 1. **Shared Experts (Global):** Dedicated to encoding **cross-domain common knowledge**. They extract fundamental visual primitives that remain invariant across different sensor modalities and datasets, ensuring consistent semantic capture.
> 2. **Dataset-Specific Experts (Static):** Tasked with capturing **domain-specific intrinsic knowledge**. They focus on unique environmental characteristics and modality signatures, effectively addressing distribution shifts.
> 3. **Dynamic Experts (Routing-based):** Shared across tasks but dynamically selected to capture **generic visual patterns (e.g., edges, basic shapes)**.
>
> **Table 1: Responsibilities of experts in HR-MoE.**
>
> | Expert Category       | Responsibility                      | Subtask Correspondence                  |
> | :-------------------- | :---------------------------------- | :-------------------------------------- |
> | Shared Experts        | Fundamental visual primitives       | Shared by $T_{1}$--$T_{5}$              |
> | Task-Specific Experts | domain-specific intrinsic knowledge | Dedicated to each dataset               |
> | Dynamic Experts       | generic visual patterns             | Dynamically routed for $T_{1}$--$T_{5}$ |
>
> We will incorporate this table and further descriptions into **Section 3.3** of the revised manuscript.
>
> ---
>
> ### **Q2: Discussion on Task-Recognition Strategy.**
>
> **A2:** Thank you for this insightful question regarding our **Task-Recognition Strategy**. We would like to clarify the inference protocol and the robust role of the **Task-Recognition Head**:
>
> * **Prior Knowledge:** In practical drone applications, the input modality (e.g., switching to IR at night) is typically known via sensor metadata. Consequently, the model can select the corresponding experts based on this prior information.
> * **Auxiliary Supervision:** The Task-Recognition Head serves as an auxiliary tool during training to "force" the backbone to learn domain-aware discriminative features. We observed that the task classification loss drops to a magnitude of $10^{-5}$, indicating that the model successfully acquires a strong capability to differentiate between various tasks and modalities.
> * **Robustness and Inference:** Even though the Task-Recognition Head is removed during inference to reduce overhead, the feature space has already been structured to be task-discriminative during training.
>
> We will add a detailed discussion on the convergence of task loss and its impact on feature decoupling in the revised version.
>
> ---
>
> ### **Q3: Performance trade-off and Model Scale Fairness.**
>
> **A3:** We follow the reviewer's constructive advice to address the fairness of comparison regarding FLOPs and parameters. We have developed two lighter versions, **DroneDINO-Tiny** and  **DroneDINO-Small**, primarily by reducing the parameter count of the backbone network and the hidden dimensions to provide a more comprehensive evaluation.
>
> **Table 2: Performance comparison across different model scales.**
>
> | Model               | Params (M) | FLOPs (G) | VisDrone(mAP) | DroneVehicle(mAP) | RGBTDronePerson(mAP) |
> | :------------------ | :--------: | :-------: | :-----------: | :---------------: | :------------------: |
> | CerberusDet         |    143     |    505    |     39.2      |       81.5        |         40.4         |
> | **DroneDINO-T**  |     52     |    346    |     45.1      |       83.5        |         53.4         |
> | **DroneDINO-S** |     74     |    382    |     45.3      |       83.6        |         52.4         |
> | **DroneDINO-L** |    225     |    656    |     50.4      |       85.5        |         55.8         |
>
> As shown in **Table 2**, both **DroneDINO-Small** and **DroneDINO-Tiny** consistently outperform the state-of-the-art CerberusDet across all three datasets, despite utilizing significantly fewer parameters and FLOPs. This empirical evidence demonstrates that our performance gains stem primarily from the architectural superiority of the **HR-MoE** design rather than mere parameter scaling. These results and the accompanying analysis will be incorporated into **Section 4.3** of the revised manuscript.

---

> > ### Author Rebuttal · Reviewer_Pc9y · 2026-04-02
> >
> > The author has addressed my concerns, I will maintain my score.

---

### Decision · Program_Chairs · 2026-04-30

**Decision:**

Accept (spotlight)

**Comment:**

This paper presents DroneDINO, a unified detection framework for heterogeneous drone perception across RGB, IR, and RGB-IR modalities. The key innovations—Heterogeneous Routed Mixture of Experts (HR-MoE) with shared, static, and dynamic expert partitioning, and a task-recognition auxiliary loss—effectively address expert collapse and negative transfer in multi-task multi-modal settings. All three reviewers provided positive assessments (weak accept, accept, strong accept) and confirmed that their concerns were fully resolved in the rebuttal, including clarifications on baseline fairness, expert balancing, efficiency analysis, and active parameter reporting. The paper is technically sound, empirically strong across multiple benchmarks, and highly relevant for deployment-oriented drone perception.